# Listeriosis Outbreak in South Africa: A Comparative Analysis with Previously Reported Cases Worldwide

**DOI:** 10.3390/microorganisms8010135

**Published:** 2020-01-17

**Authors:** Christ-Donald Kaptchouang Tchatchouang, Justine Fri, Mauro De Santi, Giorgio Brandi, Giuditta Fiorella Schiavano, Giulia Amagliani, Collins Njie Ateba

**Affiliations:** 1Department of Microbiology, North-West University, Mafikeng Campus, Private Bag X2046, Mmabatho 2735, South Africa; christdonaldk@yahoo.com (C.-D.K.T.); frijustine2000@gmail.com (J.F.); 2Department of Biomolecular Sciences, University of Urbino Carlo Bo, via S. Chiara 27, 61029 Urbino (PU), Italy; mauro.desanti@uniurb.it (M.D.S.); giorgio.brandi@uniurb.it (G.B.); giulia.amagliani@uniurb.it (G.A.); 3Department of Humanities, University of Urbino Carlo Bo, via Bramante 17, 61029 Urbino (PU), Italy; giuditta.schiavano@uniurb.it; 4Food Security and Safety Niche Area, Faculty of Natural and Agricultural Sciences, North-West University, Mafikeng Campus, Mmabatho, Mafikeng 2735, South Africa

**Keywords:** Food chain, foodborne disease, *Listeria monocytogenes*, Listeriosis, South African outbreak, sequence type 6

## Abstract

*Listeria* species are Gram-positive, rod-shaped, facultative anaerobic bacteria, which do not produce endospores. The genus, *Listeria*, currently comprises 17 characterised species of which only two (*L. monocytogenes* and *L. ivanovii*) are known to be pathogenic to humans. Food products and related processing environments are commonly contaminated with pathogenic species. Outbreaks and sporadic cases of human infections resulted in considerable economic loss. South Africa witnessed the world’s largest listeriosis outbreak, characterised by a progressive increase in cases of the disease from January 2017 to July 2018. Of the 1060 laboratory-confirmed cases of listeriosis reported by the National Institute of Communicable Diseases (NICD), 216 deaths were recorded. Epidemiological investigations indicated that ready-to-eat processed meat products from a food production facility contaminated with *L. monocytogenes* was responsible for the outbreak. Multilocus sequence typing (MLST) revealed that a large proportion (91%) of the isolates from patients were sequence type 6 (ST6). Recent studies revealed a recurrent occurrence of small outbreaks of listeriosis with more severe side-effects in humans. This review provides a comparative analysis of a recently reported and most severe outbreak of listeriosis in South Africa, with those previously encountered in other countries worldwide. The review focuses on the transmission of the pathogen, clinical symptoms of the disease and its pathogenicity. The review also focuses on the major outbreaks of listeriosis reported in different parts of the world, sources of contamination, morbidity, and mortality rates as well as cost implications. Based on data generated during the outbreak of the disease in South Africa, listeriosis was added to the South African list of mandatory notifiable medical conditions. Surveillance systems were strengthened in the South African food chain in order to assist in preventing and facilitating early detection of both sporadic cases and outbreaks of infections caused by these pathogens in humans.

## 1. Introduction

*Listeria* was reported for the first time in 1924 by Murray and colleagues due to the sudden deaths of six young rabbits and was named *Bacterium monocytogenes* [1]. Subsequently, in 1927, Pirie named the bacterium *Listerella hepatolytica* in honour of Sir Joseph Lister, a pioneer in the field of antisepsis after successfully isolating it from the liver of several gerbils (*Latera lobenquiae*) in South Africa [2]. In 1929, three patients suffered from infectious-mononucleosis, due to *Listeria* and were the first confirmed human cases in Denmark [3]. In 1940, the name *Listeria monocytogenes* was finally adopted to harmonise the nomenclature of the bacterium.

*Listeria* species are Gram-positive, catalase positive, oxidase negative and facultative anaerobic bacteria. Morphological identification presents cells as chains or single rods, non-sporulating, possessing peritrichous flagella, with a tumbling motility [4]. *Listeria* measures approximately 0.5 µm in diameter and 1–2 µm in length. The genus is classified under the family *Listeriacaeae*, which comprises 17 species, including *L. monocytogenes, L. ivanovii*, *L. seeligeri, L. innocua, L. welshimeri, L. grayi*, and the more recently characterised *L marthii, L. rocourtiae, L weihenstephanensis* and *L. fleischmannii, L. floridensis, L. aquatica, L. cornellensis, L. riparia, L. grandensis, L. booriae* and *L. newyorkensis* [5,6]. *Listeria* are capable of tolerating up to 10% salt, pH of 6–9 and temperatures of 0–45 °C with an optimal temperature range of 30–37 °C [7,8]. *Listeria* species are widespread in nature; however, among them, *L. monocytogenes* and *L. ivanovii* are human pathogens, causative agents of listeriosis [9,10]. Listeriosis caused by *L. monocytogenes* is a severe disease with a mortality rate of 20–30% [11]. This foodborne disease affects numerous individuals worldwide with a high fatality rate [1].

The pathogen is a major threat to the food industry as it can survive and proliferate in extreme conditions such as high salinity, acidity, and refrigeration temperatures [12]. Due to its ubiquitous and widespread nature, it was found in diverse natural environments such as plants, soil, silage, animals, sewage, and water. Contaminated food such as vegetables, dairy products, red meat, poultry, and seafood are implicated as sources of infection [13,14]. Sporadic and epidemic incidents of listeriosis were reported due to the consumption of ready-to-eat (RTE) foods, dairy products, seafood, pre-cooked or frozen meat, pork, and fresh produce. In the modern, fast-paced lifestyle, the population is consuming more RTE and ready-to-cook food products that entail minimal processing [15,16]. There is, therefore, a high risk of food contamination, which can cause severe clinical health effects in humans.

## 2. Potential Sources of Contamination with *Listeria* Species

*Listeria monocytogenes* is the causative agent of multiple outbreaks of listeriosis worldwide [17] associated with a variety of foods, such as milk and other dairy products, vegetables, salads, meat products and ready-to-eat foods (i.e., smoked salmon). Outbreaks caused by contaminated fishery products were also reported [18]. Poor manufacturing practices might result in food contamination by this pathogenic bacterium in the food supply chain [19]. A previous report showed that factors such as poor sanitary practices, contaminated processing environments and temperature abuse during overstay storage in retail outlets led to *Listeria*-contaminated vacuum-packed (VP) smoked fish products [20]. Numerous organisations worldwide such as World Health Organisation (WHO), Food and Agricultural Organisation (FAO) and Codex Alimentarius, among others have been applying the policy of “Zero Tolerance” for *L. monocytogenes* in RTE processed foods to reduce the high risk of food contamination and, consequently, reducing the spread of infection to the public [21,22]. *Listeria* is widely dispersed in nature and was isolated from soil, plants, sewage, and water [23]. From these sources, the bacterium can propagate, disseminate and contaminate food products, a source of infection to animals and humans. Due to the severity of the disease, stringent procedures must be established to detect the critical control point where contamination is at its maximum level. It is worthy to note that approximately 20–30% of both sporadic and outbreak *L. monocytogenes* cases are fatal [24,25].

## 3. Transmission of the Disease

The reservoirs of infection are the soil and the intestinal tracts of asymptomatic animals, such as wild and feral mammals, birds, fish, and crustaceans [26]. Infected animals can shed *L. monocytogenes* in their faeces [27], milk [28] and uterine discharges [29]. It is also found in aborted foetuses and, occasionally, in the nasal discharges and urine of symptomatic animals [30]. Soil or faecal contamination results in its presence in plants and silage [13]. Most infections are acquired through ingestion; however, *Listeria* can also be spread by inhalation or direct contact [31]. Venereal transmission might also be possible [32]. In ruminants, listeriosis typically occurs after the consumption of contaminated silage or other feed [33]. For humans, contaminated sources of food include raw meat and fish, unpasteurised dairy products and uncooked vegetables. *L. monocytogenes* has also been found in foods contaminated after processing, particularly soft cheeses, deli cold cuts, sliced or grated cheese and ice cream. The infective dose for oral transmission is unknown but it is thought to depend on the bacterial strain and the host susceptibility [34,35]. Processed food may get contaminated during the production process from the raw product to the final consumer. Due to the significant increase in the level of contamination and the risk of exposure of food to *L. monocytogenes*, food security and safety is of global concern [36]. Emphasis is now placed on the negligence of food production/processing facilities and rethinking consumer habits towards the consumption of convenient foods [37]. A major issue facing the chain of production is the fact that the *Listeria* pathogen can survive under low-temperatures, resulting in increased cases and high mortality rates due to the infection. It is worth mentioning that listeriosis due to *L. monocytogenes*, is not only prominent in humans but also life-threatening with various lethal symptoms [34,38]. Healthy people seem to consume most *Listeria*-contaminated foods without any clinical signs; however, in susceptible persons, the infective dose is probably about 10–100 organisms [39,40]. Vertical transmission is the usual source of infection in newborn human infants and ruminants; infections are transmitted either transplacentally or from an infected birth canal. Humans can also be infected by direct contact with infected animals during calving, lambing or necropsies. Cases were reported after contact with sick birds or the carcasses of asymptomatic poultry [41,42]. *L. monocytogenes* is relatively resistant to freezing, drying and heat. It will grow at temperatures from 1 °C to 45 °C and can even proliferate at refrigeration temperatures on contaminated foods. It can tolerate a pH from 3.6 to 9.5 and a sodium chloride content of up to 10%. A pH of greater than 5 (e.g., in spoiled silage) favours the growth of this organism; however, it has also been found in silage with a pH less than 4 [43,44,45].

## 4. Clinical Signs and Symptoms of Listeriosis

Listeriosis is a fatal infection that mostly affects pregnant women, newborns, the elderly and immunocompromised or debilitated hosts. In symptomatic individuals, common symptoms include diarrhoea, abdominal pain, fever, vomiting, mild and flu-like illness. Pregnant women may also experience chills, headache, slight dizziness, or gastroenteritis [46,47,48]. Sepsis (bacteraemia) in pregnant women may eventually lead to abortion, stillbirth, premature birth, or septicemia in the newborn. *L. monocytogenes* infection might result in side-effects on the central nervous system, such as encephalitis, rhombencephalitis (brainstem encephalitis), or focal signs suggestive of brain abscess formation [49]. Although common symptoms of relatively mild gastroenteritis are observed in healthy adults, it is severe in immune-compromised patients. Complication of the disease may result in meningitis, meningoencephalitis, or septicemia, with immune-suppressed individuals highly at risk [50].

### 4.1. Asymptomatic Carriage of L. monocytogenes

*Listeria* colonises at least 1–5% of healthy adults. Moreover, every year, humans experience multiple exposures to persons with transient carriage of the pathogen [47]. Research has shown that 2 out of 3 people experience episodes of faecal carriage, which takes less than 4 days and is asymptomatic. Approximately 20–25% of faecal carriage is displayed by people who come in contact with invasive *Listeria* disease [51].

### 4.2. Gastroenteritis Due to L. monocytogenes

*Listeria*-infected humans may experience gastroenteritis. There are several side-effects associated with gastroenteritis, such as fever (60–100% of cases), non-bloody diarrhoea (33–88%), arthromyalgia (20–100%) and headache (15–88%). Fever and vomiting are common in children with gastroenteritis, while diarrhoea and arthralgia are common in adults. The incubation period ranges from 6 hours to 10 days and the symptoms span from 1–3 days, although they can last for up to one week. Hospitalisation caused by gastroenteritis due to listeriosis is usually prominent in children and the elderly. Blood cultures from gastroenteritis patients may yield *Listeria* [52].

### 4.3. Listeriosis in Pregnancy

Listeriosis infection in pregnant women presents a serious side-effect because of its invasive nature. A previous study revealed the mean incubation period of listeriosis in pregnancy as 27.5 days, with a range of 17–67 days. Major side-effects include mild flu-like symptoms, with fever, backache, and headache, and some might lead to neonatal sepsis. *Listeria* infection in pregnancy usually occurs at the early stages, and the side-effects are observed during the third trimester. The disease often results in poorer neonatal outcomes. Serious side-effects associated with infections in pregnant women are spontaneous abortion, stillbirth, or preterm birth [53].

### 4.4. Neonatal Infections Due to L. monocytogenes

*Listeria* infection in neonates occurs through an endogenous transplacental transmission, inhalation of infected amniotic fluid, or following colonisation from the maternal gastrointestinal or vaginal carriage. A study revealed the isolation of *Listeria* pathogens from the genital tract and blood cultures of majority of mothers. The premature offspring experienced the onset of the disease within 36 hours, which represented transplacental neonatal infection. Neonates exhibited sepsis (90%), respiratory distress or pneumonia (40%), meningitis (25%) and, sometimes, with inflammatory granulomata (granulomatosis infantiseptica). The appearance of rash occurs with maculopapular or papulovesicular lesions on the trunk or extremities. Between 5–30 days postnatal period, the late-onset of the disease progresses and patients present the expansion of non-specific symptoms, sepsis and meningitis [54,55].

### 4.5. Bacteraemia Due to L. monocytogenes

*Listeria monocytogenes* can disseminate into the blood, leading to bacteraemia, one of the clinical presentations in some *Listeria*-infected persons. Bacteraemia due to *Listeria* might lead to gastroenteritis and, in most cases, associated with pregnancy or neonatal infection. Bacteremia in adults might be a clinical effect of HIV infection, chronic renal disease, steroid use, underlying malignancy, chemotherapy or age >65 years [56,57].

### 4.6. Meningitis Due to L. monocytogenes

Acute meningitis and encephalitis are considered to be some of the most severe symptoms of listeriosis. Other symptoms include rhomboencephalitis, which is the involvement of the midbrain, pons and/or cerebellum linked to cranial nerve involvement or cerebellar signs (ataxia, tremor), or the development of hemiparesis. The incubation period of meningitis is estimated to be 0–21 days, with an average of 10 days. Studies revealed that cases of neuroinvasive listeriosis, neck stiffness was found in 75% of cases, focal neurological signs in 30%, seizures in 30% and coma in 7%. Focal neurological symptoms comprised single or multiple cranial nerve involvement (most commonly the 6th and 7th cranial nerves), hemiparesis, ataxia, and aphasia [58,59].

## 5. Pathogenicity Factors and Virulence Potential

The pathogenicity of *L. monocytogenes* is due to the expression of different genes, which are responsible for its ability to penetrate, proliferate and spread in cells [60]. These capabilities are attributed to their *inlAB* internalisation locus, Listeria pathogenicity island-1 (*LIPI-1*), and *hpt* intracellular growth locus, respectively (Figure 1) [49]. *InlAB* internalisation locus encodes two surface proteins: *InlA* and *InlB* that bind their corresponding receptors on host cell surface through their Leucine-Rich Repeat (*LRR*) domains, i.e., *InlA* binds human E-cadherin, a calcium-dependent intercellular adhesion glycoprotein, while *InlB* binds hepatocyte growth factor receptor Met [61] and, together, mediate bacterial internalisation and invasion into host cells. *LIPI-1* encodes a pore-forming toxin listeriolysin O (*LLO*) and two phospholipases C (*PlcA* and *PlcB*), which cooperate to lyse the phagocytic vacuole membrane of host cells [62]. The hpt intracellular growth locus encodes hexose-6-phosphate transporter and actin assembly inducing protein (*ActA*), which play an important role in intracellular bacterial growth, cell-to-cell spread and actin polymerisation [49].

The virulence of *L. monocytogenes* is mainly regulated by six genes (*prfA*, *plcA*, *hly*, *mpl*, *actA* and *plcB*) residing in *PrfA*-dependent virulent gene clusters and other virulence-related genes located outside this gene cluster [49]. Furthermore, *L. monocytogenes* carries a gene cluster of five genes termed stress survival islet 1 (*SSI-1*), contributing to the survival of cells under suboptimal conditions, including low pH and high salt concentrations especially within food environments [63].

The *prfA* virulent gene clusters of *L. monocytogenes* and *L. ivanovii* also contains a positive master-regulator that controls the expression of different virulence factors. In addition, these pathogenic strains possess the *hlyA* gene that encodes a 60-kDa sulfhydryl-activated pore-forming listeriolysin O (*LLO*) that is very essential for facilitating escape of invading bacterial cells from the phagosomes of host cells into the host cytosol. The *LLO* is, therefore, known to be the main virulence feature responsible for discharging bacterial cells from the primary and secondary intracellular vacuoles [64]. The species-specificity properties of the *hlyA* gene and the *LLO* makes them effective molecular identification targets for *L. monocytogenes*, especially in food samples [65], on-farm microbiological control initiatives [66], and for serological diagnosis of ovine listeriosis [67] as well as contamination in pregnant women [68]. On the other hand, the *plcA* and *plcB* code for phosphatidylinositol-specific phospholipase C and phosphatidylcholine phospholipase respectively that assist, can operate in concert to ensure lysis of host cell membranes [69]. Soni, Ghosh [64] state that virulent genes comprising members of the internalins protein family (*inlA, inlB*), encode cell wall proteins that are important for attachment and invasion of non-phagocytic host cells. These virulent genes can be expressed or repressed by a few environmental conditions, thus affecting the virulence of *L. monocytogenes* [70].

## 6. Antibiotic Resistance in *Listeria monocytogenes*

The first *L. monocytogenes* strains resistant to antimicrobials were detected in 1988 [72]. Since then, an increasing number of strains resistant to one or more antibiotics, isolated from humans, animals, food [73], such as food processing environments [74] and dairy farms [75], were reported in recent years.

Although *L. monocytogenes* are considered vulnerable to a wide range of antibiotics, intrinsic antibiotic resistance to first generation Quinolones, Fosfomycin, Monobactams and broad-spectrum Cephalosporins were described [76]. Moreover, clinical use of antibiotics contributed to the selection of resistant strains, particularly for antibiotics commonly used to treat listeriosis. β-lactams (Penicillin and Ampicillin), with or without Gentamicin, are the main antibiotics considered for the treatment of listeriosis; Vancomycin and Trimethoprim / Sulfamethoxazole can be used as alternative therapy for Penicillin-allergic patients [77].

Veterinary use of antimicrobials in food-producing animals, commonly administered for disease therapy, prophylaxis and as growth promoters, could also contribute to the emergence of resistant strains [78]. Antibiotic resistance and particularly multi-resistance represent public health problems since they may cause failure of therapeutic treatment. Therefore, monitoring changes in antibiotic resistance of *L. monocytogenes* due to the continuing emergence of resistant strains, particularly by integration of phenotypic and genotypic techniques, is needed.

## 7. Outbreak of *Listeriosis*

### 7.1. Outbreak of Listeriosis in the United States of America

In 1998, an outbreak occurred in 10 states linked to the consumption of hot dogs and possibly refrigerated, processed deli meat. Forty patients contaminated by a single strain, *L.monocytogenes*, were identified in early August 1998 from the States of Ohio (13 cases); New York (12); Tennessee, Massachusetts and West Virginia (three each); Michigan (two); Connecticut, Oregon, Vermont, and Georgia (one each). Out of the forty infected patients, information could only be accessed from 38 individuals comprising six newborns and 32 adults (median age: 69 years; range: 18–88 years) with 55% of patients female. Out of this number, deaths were recorded in one foetus and three elderly [79]. The same year, on 22 December, Bil Mar Food recalled the production lots EST P261 or EST 6911 of hot dogs and meat products that could be contaminated. This recall comprised the Ball Park, Bil Mar, Bryan Bunsize and Bryan 3-lb Club Pack, Grillmaster, Hygrade, Mr Turkey, Sara Lee Deli Meat and Sara Lee Home Roast brands [80].

*Listeria monocytogenes* was also isolated from 29 patients in 10 states between 17 May and 26 November 2000. Out of this number, majority of cases 26 (90%) occurred from 15 July to 26 November 2000, while fewer cases 3 (10%) were isolated in the earlier months. The 29 patients were distributed as follows: New York (15 cases); Georgia (3); Connecticut, Ohio and Michigan (2 each); California, Pennsylvania, Tennessee, Utah and Wisconsin (1 each). 21 patients were elderly (29–92 years) with a median age of 65 years while 8 were perinatal. Three cases of miscarriages or stillbirths and four deaths were reported [81]. Since then, outbreaks or sporadic cases of the infection have been reported almost every year. Table 1 provides a summary of outbreaks of listeriosis in the USA from 1976–2019.

### 7.2. Outbreaks of Listeriosis in the European Union

Cases of Listeriosis in the European Union were reported in 12 countries (667 cases) in 1999, rising to 1583 cases in 23 countries in 2006; a total of 667 cases were reported in 12 countries in 1999, 586 in 2000 (13 countries), 872 in 2001(13 countries), 909 in 2002 (15 countries), 1070 in 2003 (16 countries), 1264 in 2004 (22 countries), 1427 in 2005 (19 countries) and 1583 in 2006 (23 countries). This report shows that human cases of listeriosis are on the increase each year. Table 2 shows the numbers cases reported for Listeriosis in human beings in the EU from 1999–2006,

Based on the definition by the European of cases of outbreak, as of 6 March 2018, a multi-country foodborne outbreak was verified in five countries, involving 32 confirmed cases and six deaths due to or with the infection (Table 3) [104].
(1)Austria reported two confirmed cases from 2016. Both cases were males, aged over 85 years at symptom onset, one case was fatal.(2)Denmark reported four confirmed cases with sampling dates in January 2017, May 2017, and February 2018 (two cases). Three cases were female patients and one male. Their ages ranged between 37 and 74 years while one case was fatal [105].(3)Finland reported 14 confirmed cases with sampling dates from September 2016 to January 2018. Nine cases were females and five males. Their ages ranged between 22 and 92 years while two cases were fatal.(4)Sweden reported six confirmed cases with isolates. Out of this number, five were females while one was a male with an age range between 70 and 94 years. Two cases were fatal.(5)The United Kingdom reported six confirmed cases between 2015 and 2018 with an age range of 22–84 years. Four cases were males and two females.

### 7.3. Outbreaks of Listeriosis in Australia

*Listeria monocytogenes* has become an invasive pathogen in foods and processed food products. In 2003, Australia experienced an outbreak of the disease with the highest mortality (30%) [99]. In 2009, there was contamination of chicken wraps sold on domestic flights while in 2010, *L. monocytogenes* contaminated melons and caused a multijurisdictional outbreak, with approximately 70 cases reported (Figure 2). The highest number of cases of Listeriosis was reported in 2014 and the most prevalent group affected were patients of 80 years and above (Figure 2). In 2018, the isolated *L. monocytogenes* strain was serotype 4b strain ST240 and affected 59% of females. 

### 7.4. Outbreaks of Listeriosis in Asia

In the past few decades, Asia has experienced severe outbreaks of Listeriosis. Several reports in Asian countries such as Japan [106,107], Thailand (Bangkok) [108] and Taïwan [109,110] were documented for clinical cases of Listeriosis. Table 4 shows that *L. monocytogenes* targets immune-compromised individuals such as pregnant women and infants. This is displayed in reports of outbreaks in China in 1964 and 2010 [111].

### 7.5. Outbreaks of Listeriosis in Nigeria

Nigeria is considered to be the "Giant of Africa" due to the large population and growing economy. Regardless of this, there have been limited reports documented and surveillance systems of outbreaks of Listeriosis in the country. However, researchers successfully isolated *Listeria* in humans, animals, the environment, and food samples in Nigeria (Table 5). Listeria was shown to affect pregnant women and infants while ruminants dominated reports of incidences in animals.

### 7.6. Outbreaks of Listeriosis in South Africa

Currently, South Africa is experiencing severe outbreaks of Listeriosis. To date, this is the highest outbreak of *Listeria* in history affecting a single nation. A total of 820 cases of Listeriosis were reported by the National Institute of Communicable Diseases (NICD) from January 2017 to 23 January 2018. The highest number of cases originated from Gauteng Province (59%), followed by the Western Cape (13%) and Kwa Zulu-Natal (7%). Samples were collected from blood (71%) and Cerebral Spinal Fluid (CSF) with 23% in the public and private healthcare sectors. Approximately 66% were diagnosed from public and 34% from private healthcare sectors. The age distribution of cases ranged from birth to 93 years, with majority (96%) of cases neonates (≤28 days). The gender distribution was 55% females and 45% males [142]. Figure 3 shows the evolution of confirmed cases of Listeriosis from 1 January 2017 to 17 July 2018 in South Africa [23].

Fortunately, human Listeriosis is rare. However, there is a high case of fatality if the disease ensues. With the very long incubation period, which can range up to 60 days, attributing the illness to a specific food product becomes difficult [17]. In addition to several factors mentioned earlier, responsible for outbreaks of Listeriosis in the supply chain, resistance of microorganisms to antibiotics and disinfectants also play a role [143].

### 7.7. Estimated cost of the outbreak of Listeriosis in South Africa 

Food spoilage caused by microbial pathogens results in massive economic loss to both the company responsible and the country. In developing countries, such as South Africa, outbreaks of Listeriosis are poorly documented [23], including the major outbreak in 2017–2018, which caused a deadly epidemic. Approximately, 1060 cases were reported with 216 deaths [23]. Processed meat products (Polony), produced by Tiger Brands Limited (South Africa), was identified as the responsible source. The accountable factories were Tiger Brand (Polokwane, Limpopo and Germiston, Johannesburg) [144]. The economic loss was projected to be US $260 million (R 3, 79 Billion) in terms of lethality, while cost of hospitalisation associated with one-month recovery of affected patients by the disease was US $10.4 million (R1, 5 Billion). Furthermore, more than US $15 million (R2, 19 Billion) was attributed to loss due to productivity and export of food processing companies due to the outbreak [145].

## 8. Conclusions

It is confirmed in this review that South Africa recorded the highest outbreaks of Listeriosis with a high mortality rate. *L. monocytogenes* was successfully isolated from contaminated possessed meat (Polony). South Africa is considered the economic capital of Africa and developing exponentially. The country also has strong regulations on food safety management and laws aligned with international standards. Despite this, the recent outbreak of Listeriosis showed some limitations in food safety policies. Authorities in the Department of Health (South Africa) recently reviewed the list of notifiable diseases to include Listeriosis. 

## Figures and Tables

**Figure 1 microorganisms-08-00135-f001:**
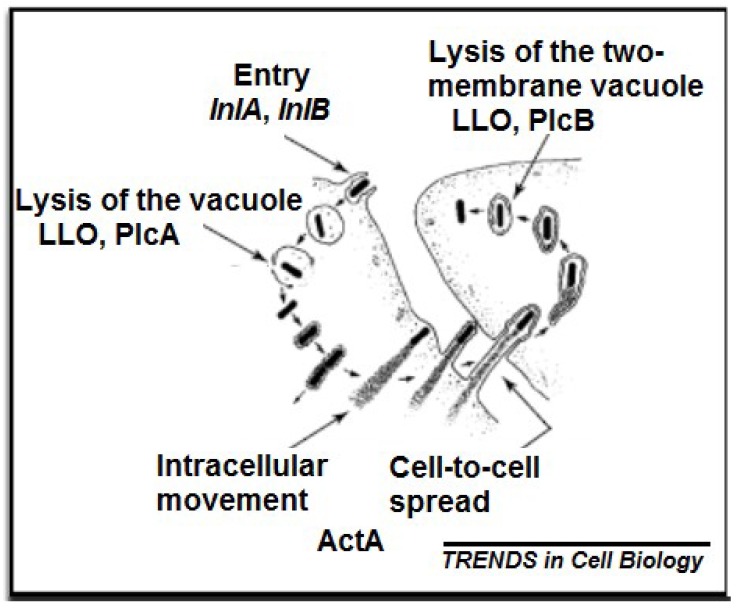
Infection process of a host cell by *L. monocytogenes* [71].

**Figure 2 microorganisms-08-00135-f002:**
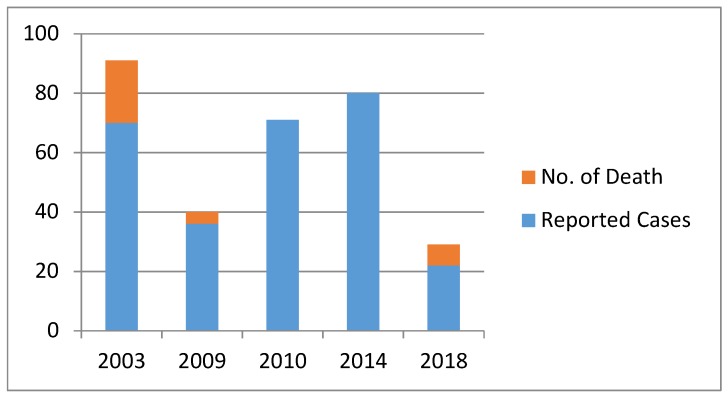
Outbreak of Listeriosis in Australia in 2003, 2009, 2010, 2014 and 2018.

**Figure 3 microorganisms-08-00135-f003:**
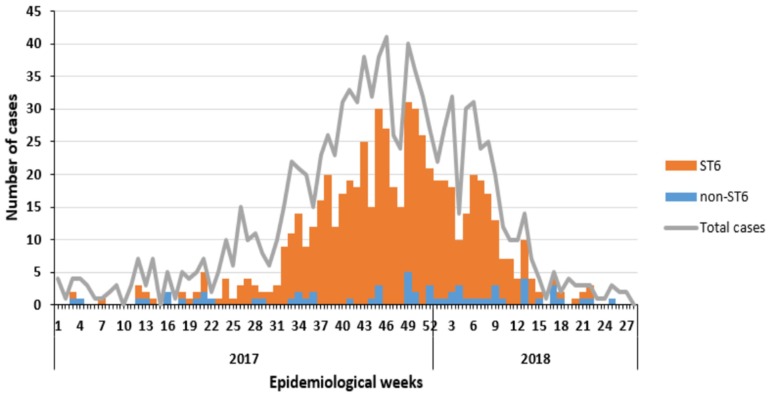
Epidemic curve of laboratory-confirmed cases of Listeriosis according to date of collection of clinical specimen (*N* = 1060) and type of sequence (ST) (*n* = 636), South Africa, 1 January 2017 to 17 July 2018 [23].

**Table 1 microorganisms-08-00135-t001:** Outbreaks of Listeriosis in the USA, 1976–2019.

*Listeriosis* Outbreaks in the USA 1976–2019			
Year	Number of Cases (Number of Deaths and Number of Miscarriages)	Food	References
1976	20	Raw salad	[31]
1983	49	Milk	[82]
1985	142 (41)	Soft cheese	[83]
1989	9 (1)	Shrimps	[84]
1998–1999	108 (14; 4)	Hot dog	[85]
2000	29 (4; 3)	Turkey meat	[81]
2001	12	Mexican style cheese	[86]
2002	46 (7; 35)	Fresh and frozen ready-to-eat turkey and chicken	[87]
2007	311 (52)	Milk	[88]
2007	5 (3)	Dairy milk	[89]
2010	10 (5)	Pre-cut celery	[90]
2011	147 (33)	Cantaloupe	[40]
2012	22 (4)	Ricotta salata Cheese	[91]
2014	5 (2)	Mung bean sprouts	[92]
2014	4	Stone fruit	[93]
2010–2015	10 (3)	Ice cream	[94]
2014–2015	5 (3)	Caramel apples	[95]
2016	9 (3)	Frozen vegetables	[96]
2016	2 (1)	Raw milk	[97]
2016	19 (1)	Packaged salads	[98]
2017	8 (2)	Soft Raw Milk Cheese Made by Vulto Creamery	[99]
2018	4	Pork Products	[100]
2018	4 (1)	Deli Ham	[101]
2019	8 (1)	Deli-Sliced Meats and Cheeses	[102]

**Table 2 microorganisms-08-00135-t002:** Cases of Listeriosis reported among humans in Europe, 1999–2006 [103].

Country		Number of Confirmed Cases						
	1999	2000	2001	2002	2003	2004	2005	2006
Austria	13	14	9	16	8	19	9	10
Belgium	64	48	57	44	76	70	62	67
Cyprus								1
Czech Republic						16	15	78
Denmark	44	39	38	28	29	41	46	56
Estonia	1					2	2	1
Finland	46	18	28	20	41	35	36	45
France	275	261	187	218	220	236	221	290
Germany	31	33	216	240	256	296	510	508
Greece	1	2	3	5		3		6
Hungary						16	10	14
Ireland		7	7	6	6	11	11	7
Italy	17	13	31			25	51	51
Latvia		36		16	8	5	3	2
Lithuania					2	1	2	4
Luxembourg								4
Malta								0
Netherlands			16	32	52	55	96	64
Poland				31	5	10	22	28
Portugal						38		
Slovakia				7	6	8	5	12
Slovenia					6	1		7
Spain	32	35	57	49	52	100	68	78
Sweden	27	46	67	39	48	44	35	42
United Kingdom	116	115	156	158	255	232	223	208
EU Total	667	586	872	909	1070	1264	1427	1583
Bulgariax		2000						6
Iceland		14						
Liechtenstein		48						
Norway			18	17	18	21	14	27

**Table 3 microorganisms-08-00135-t003:** Cases of Listeriosis verified in five countries in Europe between 2015 and 2018.

Country	Confirmed Cases (Number of Deaths)				Total Number of Cases	Total Number of Deaths
	2015	2016	2017	2018		
Austria	0	2 (1)	0	0	2	1
Denmark	0	0	2	2 (1)	4	1
Finland	0	3	10 (2)	1	14	2
Sweden	0	3 (1)	3 (1)	0	6	2
United Kingdom	1	2	2	1	6	0
**Total**	**1 (0)**	**10 (2)**	**17 (3)**	**4 (1)**	**32**	**6**

**Table 4 microorganisms-08-00135-t004:** Reporting periods of clinical cases of Listeriosis for each subgroup in China, 1964–2010.

Reporting Period	No. of Cases				
	Immune-Competent Patients	Immune-Compromised Patients	Pregnant Women	Neonates	Total
1964–1970	1	0	0	1	2
1971–1980	1	0	0	0	1
1981–1990	8	2	0	0	10
1991–2000	12	7	5	9	33
2001–2010	27	12	25	37	101
Total	48	21	30	47	147

**Table 5 microorganisms-08-00135-t005:** Different sources of *Listeria* in Nigeria, such as humans, fish, water, animals, soil, and the environment [112].

Source	Origin	References
**Human**	Patients with meningitis, septicaemia	[113,114,115]
	Neonatal and mother	[116]
	Neonates	[117]
	Blood samples	[118]
	Faecal specimens	[119]
**Animals**	Cow faeces	[120]
	African buffalo	[121]
	Guinea pig	[122]
	Pig, dog, cattle, goat, horse, camel, chicken	[123,124]
	Animals dropping	[125]
	Raw cattle milk	[126]
	Raw goat milk	[127]
	Domestic cats	[128]
	Cockroach	[129]
**Environment**	Butcher’stable	[130]
	Lake	[131]
	Irrigation water	[132]
	Nairacurrency notes	[133]
	Veterinary surgical material	[134]
	Soil	[135]
**Food**	Fish	[136]
	Vegetables	[137]
	Dried beef *Kilishi*	[138]
	*Kunu* drink	[139]
	*Wara* soft cheese	[140]
	Fermented food (*Gari* etc…)	[140]
	Frozen poultry	[141]

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
