# Peer review of "Listeriosis Outbreak in South Africa: A Comparative Analysis with Previously Reported Cases Worldwide"

_microorganisms, 2020, doi:10.3390/microorganisms8010135_

Round 1

Reviewer 1 Report

Kaptchouang Tchatchouang et al. have reviewed the current state of the Listeriosis field including new information on the outbreak in South Africa.

In general, the review is comprehensive and well written.  I have several suggestions to improve it however and some minor edits to suggest.

The section on virulence factors on Page 5 (lines 6 – 45) needs to be edited and/or rewritten to make it more concise and to correct the English.  In fact, this reviewer suggests shortening it because there are two lengthy discussions of ActA, LLO, PlcA and PlcB which are quite repetitive and mainly studied in the 1990s and not as many details on the South African Outbreak. It seems more interesting and timely to increase the length of and information in the section on the South African outbreak. 

Page 4 Line 11

Should be “results in poorer” instead of “results to poorer”

“Serious side effects associated with infection of pregnant women are” instead of “Serious side effects associated with pregnant women is”

Page 4 Line 31

Shoud be “have been regarded as one of the most severe” instead of “one of the severe”

Page 5 Line 18

Should be “anti-LLO antibody detection” instead of “antibodies detection”

Page 5 Line 29

Should be “operate in concert” instead of “operate supportively”

Should be “These virulence genes” instead of “These virulent genes”

Page 5 Line 34

Should be “Using the ActA” instea of “Using ActA”

Page 5 Line 36

Should be “into the host cytoplasm, and co-opts proteins of the host cytoskeleton”

Page 10 Line 7

Should be “To date” instead of “Till date”

Page 11 Line 12

Should be “listeriosis outbreaks are poorly” instead of “is”

Author Response

Reviewer 1: Comments

Response

The section on virulence factors on Page 5 (lines 6 – 45) needs to be edited and/or rewritten to make it more concise and to correct the English. 

In fact, this reviewer suggests shortening it because there are two lengthy discussions of ActA, LLO, PlcA and PlcB which are quite repetitive and mainly studied in the 1990s and not as many details on the South African Outbreak. It seems more interesting and timely to increase the length of and information in the section on the South African outbreak. 

The section on virulence factors on Page 5 (lines 6 – 45) has been edited and rewritten and thus occurs in a more concise manner.

The whole manuscript has been Language edited and has undergone English Language check. 

The section ActA, LLO, PlcA and PlcB has been shortened and all repetitive sections have been corrected.

Page 4 Line 11

Should be “results in poorer” instead of “results to poorer”

“Serious side effects associated with infection of pregnant women are” instead of “Serious side effects associated with pregnant women is”

Page 4 Line 11

The words “results to poorer” has been changed to “results in poorer”.

The sentence “Serious side effects associated with pregnant women is” has been changed to “Serious side effects associated with infection of pregnant women are”.

Page 4 Line 31

Shoud be “have been regarded as one of the most severe” instead of “one of the severe”

Page 5 Line 18

Should be “anti-LLO antibody detection” instead of “antibodies detection”

Page 5 Line 29

Should be “operate in concert” instead of “operate supportively”

Should be “These virulence genes” instead of “These virulent genes”

Page 5 Line 34

Should be “Using the ActA” instea of “Using ActA”

Page 5 Line 36

Should be “into the host cytoplasm, and co-opts proteins of the host cytoskeleton”

Page 10 Line 7

Should be “To date” instead of “Till date”

Page 11 Line 12

Should be “listeriosis outbreaks are poorly” instead of “is”

Page 4 Line 31

The sentence “one of the severe” has been changed to “have been regarded as one of the most severe”.

Page 5 Line 18

The words “antibodies detection” has been changed to “anti-LLO antibody detection”

Page 5 Line 29

The words “operate supportively” have been changed “operate in concert”.

The words “These virulent genes” have been changed to “These virulence genes”.

Page 5 Line 34

The words “Using ActA” have been changed “Using the ActA”.

Page 5 Line 36

The sentence now appears as “into the host cytoplasm, and co-opts proteins of the host cytoskeleton”

Page 10 Line 7

Should be “To date” instead of “Till date”

Page 11 Line 12

The word “is” has been deleted and the sentence now appears as “listeriosis outbreaks are poorly”.

Reviewer 2 Report

This review examines recent outbreaks of listeriosis, especially in South Africa, where it continues to be a serious infection problem. The cut of the article is very much in line with public health topics, while the physiology and microbiology of Listeria strains is poorly presented. Since the major pathogenic strain in South Africa seems to be ST6 (cf. Fig. 3), one (and this Reviewer in particular) would like to see a phylogenetic tree showing the taxonomic position of this strain vs. that of other strains and also environmental, non pathogenic Listeria species.

Another problem with the article is the poor quality of Fig. 1, which seems to be taken (with permission?) from a published article. There seem to be membrane nanotubes through which Listeria would be transmitted inter-cellularly - what is the evidence and reference for that? Surerly, the Authors could  produce a nicer figures showing the cellular paths of Listeria infection of mammalian cells.

The English style is not fluent in several places, including the Abstract ( e.g. line 30) and section 1 - the acronym RTE is repeated at line 24 and 26, p. 2, for example. Absence of proper punctuation, or wrong positioning of commas is recurrent in the above sections and elsewhere.

Author Response

Reviewer 2: Comments

Response

General comments:

This review examines recent outbreaks of listeriosis, especially in South Africa, where it continues to be a serious infection problem. The cut of the article is very much in line with public health topics, while the physiology and microbiology of Listeria strains is poorly presented. Since the major pathogenic strain in South Africa seems to be ST6 (cf. Fig. 3), one (and this Reviewer in particular) would like to see a phylogenetic tree showing the taxonomic position of this strain vs. that of other strains and also environmental, non-pathogenic Listeria species.

Another problem with the article is the poor quality of Fig. 1, which seems to be taken (with permission?) from a published article. There seem to be membrane nanotubes through which Listeria would be transmitted inter-cellularly - what is the evidence and reference for that? Surely, the Authors could  produce a nicer figures showing the cellular paths of Listeria infection of mammalian cells.

The English style is not fluent in several places, including the Abstract ( e.g. line 30) and section 1 - the acronym RTE is repeated at line 24 and 26, p. 2, for example. Absence of proper punctuation, or wrong positioning of commas is recurrent in the above sections and elsewhere.

The article has been re-written.

Fig. 1 has been edited to improve its quality and contains a valid reference indicating the article from where it was obtained.

A trained English Language editor has edited the article for Language.